# Infection Route of *Parvimonas micra*: A Case Report and Systematic Review

**DOI:** 10.3390/healthcare10091727

**Published:** 2022-09-08

**Authors:** Kai Shimizu, Yuta Horinishi, Chiaki Sano, Ryuichi Ohta

**Affiliations:** 1Community Care, Unnan City Hospital, Unnan 699-1221, Japan; 2Department of Community Medicine Management, Faculty of Medicine, Shimane University, Izumo 693-8501, Japan

**Keywords:** bacteremia, case report, infection route, monobacterium infection, *Parvimonas micra*, sepsis, systematic review

## Abstract

*Parvimonas micra* (*P. micra*), a bacterium that colonizes the gastrointestinal tract, is often isolated from periodontitis and abscesses as part of a complex bacterial infection. However, reports of monobacterium infections due to *P. micra* are limited. Here, we report a case of monobacterial bacteremia caused by *P. micra* with the aim of identifying the source of the invasion and clarifying the clinical features. A 54-year-old patient presented with bacteremia due to *P. micra* and with an oral invasion that we suspected resulted from prior dental treatment. Using PubMed and Google Scholar databases, we undertook a systematic review of monobacteremia caused by *P. micra*. We identified 26 patients (mean age, 70.15 years) in our systematic review. *P. micra* bacteremia and its associated phenotypes were most frequently identified in spinal discitis, followed by epidural and lumbar abscesses, and infective endocarditis. The major risk factors were malignancy, diabetes mellitus, and post-arthroplasty. When *P. micra* is detected in blood cultures, evaluation and intervention for oral contamination may be indicated.

## 1. Introduction

*Parvimonas micra* (*P. micra*) inhabits the oral cavity and intestinal tract in humans [1]. It is an obligate anaerobe and a small (0.3–0.7 μm) Gram-positive coccus that causes various infections in immunocompromised hosts. Originally, it was identified as *Peptostreptococcus micros* in 1933, but was reclassified as *Micrcomonas micros* in 1999, and finally as *P. micra* in 2006 [2]. *P. micra* infections have increased in nosocomial and immunocompromised hosts because of aging societies [2]. Primary care healthcare workers need to be aware of these types of infections.

*P. micra* is a component of the oral, upper respiratory, and intestinal microflora, and is of major clinical and bacteriological importance owing to its high isolation rate within clinical material, owing to poor hygiene among immunocompromised older patients in aging societies [2]. Its pathogenic potential has been implicated in chronic periodontal disease, alveolar abscess, peritonsillar abscess, chronic sinusitis, chronic otitis media, and pulmonary pyogenic disease [3]. It is known to be involved in deep-seated infections, such as those occurring around artificial joints [3]. The pathogenicity of *P. micra* includes the presence of capsid, high protease activity, and hydrogen sulfide toxicity produced by the utilization of glutathione [3]. Socially isolated patients, those with a low socioeconomic status, those with poor hygiene, and immunocompromised patients are vulnerable to this infection and may be at high risk owing to the absence of habitual oral care [2,3]. 

*P. micra* is difficult to identify because of its lack of clinical symptoms and slow growth, and the need for special culture media and identification methods [4,5,6]. Pyogenic spondylitis is the most common infection caused by *P. micra*, and pyogenic arthritis (post-knee arthroplasty), infective endocarditis, pleurisy, meningitis, and brain abscess have been reported [7]. As *P. micra* is an anaerobic inhabitant of the oral microbiome, risk factors include dental procedures, periodontitis, tooth extractions, and oral infections, such as abscesses or caries on the lingual apex [7]. *P. micra* may be multi-drug-resistant and may co-infect with other multi-drug-resistant bacteria in relation to cephalosporins and quinolone, resulting in a polymicrobial etiology for endogenous oral infections, such as periodontitis, while also being detected in soft tissues, skin infections, and various abscesses [8,9,10,11]. *P. micra* co-pathogens associated with polymicrobial infections include *Streptococcus*, *Bacteroides*, and *Fusobacterium* [12]. In increasingly aging societies, healthcare workers encounter various symptoms in older patients, and the etiologies of their symptoms could derive from the bacteremia of *P. micra.* Therefore, healthcare workers should be mindful of the likelihood of *P. micra* infections among older adults.

It remains unknown how *P. micra* monoinfections are transmitted and which patient populations are affected. Clarification of the transmission of *P. micra* monoinfections is vital for prevention and effective treatment. When *P. micra* is detected in blood cultures, early diagnosis and therapeutic interventions are possible. If the most common routes of infection could be identified, along with the background of patients susceptible to *P. micra* bacteremia, clinicians could more accurately determine a targeted treatment pathway against the primary disease. However, clinical diagnosis of *P. micra* monoinfections and treatment remain challenging. Here, we present the case of a 54-year-old man with *P. micra* bacteremia. We aimed to identify the source of the invasion and its clinical features, to highlight challenges in diagnosis and treatment. We also aimed to investigate the transmission of *P. micra* monoinfections and identify the affected patient population through undertaking a systematic review of relevant case reports concerning *P. micra* bacteremia.

## 2. Materials and Methods

### 2.1. Case Report 

A 54-year-old Japanese male visited our hospital complaining of persistent fever and headaches. The headaches were mild and pulsated from the anterior to the posterior region of his head. He had a long history of dental treatment, and all of his teeth had been replaced with dentures. He had also been experiencing pain in his mouth for three days: pain was present in his mandible whilst chewing, although there was no tenderness. He had a past medical history of hypertension and stroke and was taking antiplatelet and antihypertensive medication. He was a construction worker, lived with his elderly mother, had smoked 40 cigarettes a day for 40 years, and had no history of alcohol consumption.

The patient’s vital signs on arrival were as follows: Glasgow Coma Scale, 15 points; blood pressure, 129/94 mmHg; pulse, 121 beats/min; respiratory rate, 18 breaths/min; SpO_2,_ 96% (room air); and body temperature, 39.4 °C. In the physical examination, no obvious physical abnormality was noted, such as nuchal rigidity, jolt accentuation, tenderness in the sinuses or in the head and neck, moist rales, obvious abdominal findings, costovertebral angle tapping pain, or arthritis. To exclude oral infection, sepsis, and abscess, blood tests, urinalysis, and a computed tomography (CT) scan were performed. The blood examination results showed an elevated white blood cell count, but no abnormalities were noted in the CT scan and urinalysis. The patient was sent home after blood cultures were taken on the same day. The blood culture test results were positive for *P. micra*. The patient was urged to visit the outpatient clinic, but he did not attend. In the meantime, antipyretic analgesics were prescribed, but no antibiotics. The patient attended an outpatient clinic two weeks later in good condition.

### 2.2. Systematic Review

#### 2.2.1. Search Strategy

The following keywords were searched on Google Scholar and PubMed online databases: “*Parvimonas micra*” and “bacteremia.”

#### 2.2.2. Inclusion and Exclusion Criteria

Studies written in Japanese and English and published after 2006 on bacteremia associated with *P. micra* and associated diseases, such as sepsis, abscess, and spondylitis, were included. Studies published prior to 2006, those written in languages other than Japanese or English, and those in which *P. micra* was detected in samples other than blood were excluded. Inclusion and exclusion criteria are summarized in Table 1.

#### 2.2.3. Data Extraction

We used the search terms “*Parvimonas micra*” and “bacteremia” in Google scholar and PubMed online databases. The first and second authors read the full text and independently extracted the data. After data extraction, they discussed the extracted data, and once consensus had been reached regarding the inclusion of a study, the data were captured in an Excel file. This systematic review was registered in PROSPERO (registration number: 316707) and conducted according to standard procedures.

#### 2.2.4. Analysis

Quantitative and qualitative data are presented as descriptive statistics. Collected data were divided into various categories.

## 3. Results

### 3.1. Search Results

In total, 45 studies met the inclusion criteria. Of these, seven studies were excluded because of duplication. After reviewing the entire texts, 15 studies were excluded for the following reasons: two studies had been written in Korean or German, and in 13 studies, the samples used for detection were not blood samples. Finally, a total of 23 studies were identified, and 26 case reports were included in the final analysis (Figure 1) [4,11,13,14,15,16,17,18,19,20,21,22,23,24,25,26,27,28,29,30,31,32,33]. 

### 3.2. Demographics

The mean age of the patients with *P. micra* bacteremia was 70.15 years (age range: 41–94). Males were the predominant sex (80.8%). The most common complaints were fever (*n* = 13, 50%), weight loss (*n* = 3), malaise (*n* = 3), and nausea and vomiting (*n* = 2). In addition, back pain was the chief complaint in all eight cases of spondylodiscitis and epidural abscess. In the case of septic arthritis of the knee, tenderness of the right knee was the chief complaint.

### 3.3. Medical History

Of 26 patients, the underlying diseases were spondylodiscitis in seven (26.9%) patients, epidural abscess in five (19.2%) patients, psoas abscess in four (15.3%) patients, infective endocarditis in four (10.5%) patients, and malignancy in three (11.5%) patients. The most frequent comorbidity was the history of diabetes mellitus in eight (30.7%) patients, hypertension in seven (26.9%) patients, malignant tumor in three (11.5%) patients, dyslipidemia in two (7.69%) patients, and artificial joint replacement in two (7.69%) patients.

### 3.4. Entry/Route of Infection

There were 13 reported cases of oral infection (50%), one respiratory tract infection (3.84%), one bacterial translocation (3.84%), one mucosal infection (3.84%), and one percutaneous infection (3.84%). Nine (34.6%) patients had no recorded details regarding the route of infection.

### 3.5. Outcome

Following the hospitalization period, 21 of 26 (81.8%) patients survived (Table 2).

## 4. Discussion

We performed a systematic review to identify the route of infection of *P. micra* and its clinical features. Spinal discitis, followed by epidural abscess, lumbar abscess, and infective endocarditis, were most frequently associated with *P. micra* bacteremia. The most frequent risk factors were malignancy, followed by diabetes mellitus and post-prosthetic joint replacement. We found that oral infections due to dental treatment and dental caries were the most frequent causes, although many cases were reported to have an unknown route of infection.

No specific symptoms related to *P. micra* infection were observed. Fever was present in 46% of the reported cases, and other chief complaints included general malaise, impaired consciousness, and anorexia. Pain was the main complaint in all patients, which was associated with inflammation of the musculoskeletal system, such as spondylitis and arthritis. In patients with pain, an aggressive search for the source of infection, such as local abscess formation, is necessary.

It is currently unclear why the spinal region is the primary target for these infections [29]. In a case series of spondylitis due to *P. micra* reported by Durovic et al. [28], even a mild fever and mildly elevated inflammatory response resulted in a positive blood culture rate comparable to that detected in a CT-guided biopsy. This suggests that anaerobic infection by *P. micra* should be suspected even in the absence of symptoms.

All patients who presented with infective endocarditis had a history of dental procedures, such as tooth extractions. It has long been known that transient bacteremia due to dental procedures is a cause of infective endocarditis [34]. Bacteria entering the bloodstream after dental procedures, such as tooth extraction, are rapidly eliminated by the liver and other reticuloendothelial tissues, and most disappear from the bloodstream within a short period; this is referred to as transient bacteremia [35]. Sakamoto [36] reported that the frequency of bacteremia after tooth extractions was 100%, with wisdom tooth extractions accounting for 55%, and that the frequency of bacteremia after tartar removal was 70%. The frequency of bacteremia has been reported to be higher for periodontal extractions [37]. Non-dental bacteremia has been reported as being due to daily tooth brushing (30%), chewing (38%), and dental flossing [38,39,40,41,42,43,44]. Of these, the risk associated with tooth brushing has been emphasized in recent years [38,39]. The presence of bacteremia induced in the oral cavity through such routine practices has led to a strong recognition of the importance of routine oral follow-ups [39,45]. The European Society of Cardiology guidelines recommend that, as a general precaution, high- and intermediate-risk patients should undergo rigorous dental follow-ups twice a year for if they are at particularly high risk and once a year otherwise. To prevent infectious endocarditis, control of periodontal disease and caries is critical, and regular dental and oral management is crucial, even when a patient is asymptomatic.

Given *P. micra* forms part of the oral microflora, the main risk factors and routes of infection for *P. micra* bacteremia are dental treatments such as periodontitis, tooth extractions, and dental abscesses [46]. In our systematic review, 13 (50%) patients were identified as having experienced at least one of these complications. The pathogenicity of *P. micra* in oral infections has been attributed to factors such as adhesion to gingival epidermal cells, cell morphology proteolytic enzymes, and the macrophage response [47,48]. Most patients had received dental care prior to infection. In this study, other relevant patient histories included type 2 diabetes mellitus, colorectal cancer, pressure ulcers, and aspiration. Previous studies have reported that *P. micra* bacteremia is associated with aspiration pneumonia, suggesting that postoperative stress and increased dead space due to increased sputum production may have led to an increase in *P. micra* [16,17,19]. In addition, some cases of *P. micra* bacteremia possibly due to ulcers or endoscopic procedures have been reported, and trauma due to intestinal mucosal injury may predispose one to bacteremia. Nine cases had no description of the route of infection in the case presentation or discussion; these patients presented with liver abscess, splenic abscess, spinal discitis, epidural abscess, intradural abscess, and diabetes mellitus, and had no history of dental treatment.

The route of infection in one (3.84%) patient was considered to be bacterial translocation. Several factors contribute to the development of bacterial translocation. Most are abnormalities in the intestinal microbiota, disruption of the physical and immune barrier of the gut, and a reduction in systemic immunity [49]. For example, cases involving intestinal obstruction, jaundice, inflammatory bowel disease, malignancy, emergency surgery, and gastric colonization with microorganisms have been reported [50,51,52,53,54,55,56,57,58,59,60,61,62]. In our patient, malignancy may have compromised host immunity and intestinal barrier function, and the bacteria may have therefore metastasized. Therefore, it is necessary to search for immunosuppression or gastrointestinal malignancy as a possible cause of *P. micra* bacteremia.

Concerning treatment and antimicrobial susceptibility, in one previous study, 3.2% of patients did not respond to treatment with metronidazole [63]; however, most patients did respond to treatment with common antimicrobial agents. Although *P. micra* is considered to have good antimicrobial susceptibility, drug selection should be monitored through susceptibility testing [62,63]. When an abscess forms, surgical drainage is essential because antimicrobial agents rarely migrate into the abscess [52,53,54,55]. Our review showed that *P. micra* caused abscesses in 38.4% of patients. Drainage was performed in almost all cases involving liver abscesses, splenic abscesses, pyothorax, epidural abscesses, iliolumbar muscle abscesses, pyogenic pericarditis, infected knee arthritis, and brain abscesses, in addition to five out of seven cases involving splenic abscesses. Control of the infected site through drainage and antibiotic therapy is important in cases of abscess formation [52,53,54,55].

Regarding prognosis, one patient with in situ carcinoma became debilitated owing to the infection and ruptured an abdominal aortic aneurysm, which ultimately resulted in death [57,58,59]. Anaerobic infections by bacteria such as *P. micra* require early detection and treatment [53,54,55,56]. In patients who are at risk of severe disease, the threshold for searching for infections such as abscesses and infective endocarditis should be lowered for those with diabetes mellitus, non-invasive cancer, and poor oral hygiene, even in those with non-specific symptoms such as fever and malaise [60,61,62]. Abscess formation should be considered, especially when accompanied with localized pain, as *P. micra* is an anaerobe and tends to cause abscess formation.

Our patient was treated in the emergency room, and the details of the disease’s course remain unknown. Our patient had a history of dental treatment, oral contamination, a painful mouth, and no other febrile findings, suggesting a high probability that *P. micra* bacteremia occurred via the oral cavity. Our patient was an immunocompetent individual with no history of diabetes mellitus, cancer, or steroid use. Based on the clinical course, the patient had transient bacteremia caused by *P. micra* that resolved spontaneously. Therefore, consistent follow-up and health maintenance are needed because of the possibility of abscess formation, infective endocarditis, or spondylitis.

## 5. Conclusions

The spine is the most common site of underlying disease for *P. micra* bacteremia, followed by epidural abscesses, lumbar abscesses, infective endocarditis, and malignancy. The presumed route of entry is oral in most cases. The major risk factors are malignancy, diabetes mellitus, and post-arthroplasty. Patients with local abscess infection experience local tenderness. Therefore, at-risk patients should undergo thorough examination and treatment for suspected abscess formation, based on physical findings, such as the presence of oral contamination and local tenderness. Bacterial translocation is one possible mechanism of bacteremia caused by *P. micra*. A malignant tumor of the gastrointestinal tract may be the underlying cause, and the possibility of tumors should be eliminated.

## Figures and Tables

**Figure 1 healthcare-10-01727-f001:**
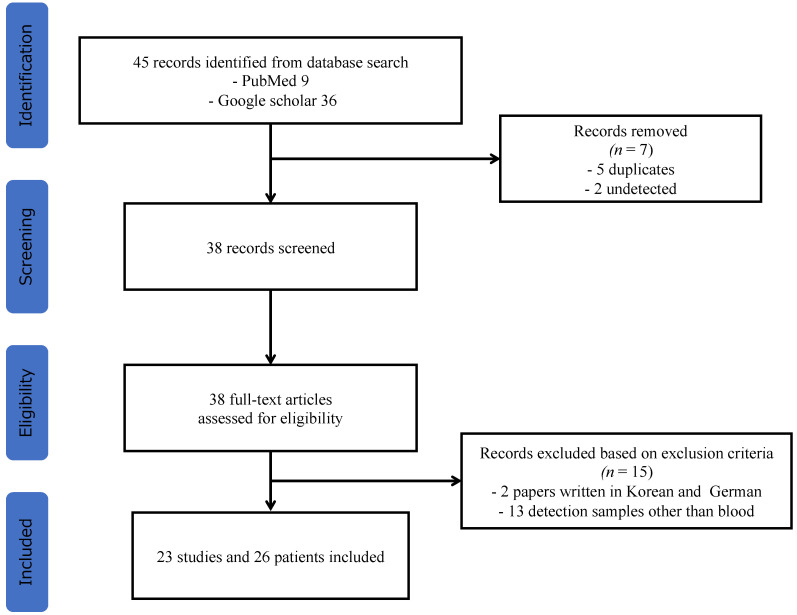
Flowchart of the study selection process for the systematic review.

**Table 1 healthcare-10-01727-t001:** Inclusion and exclusion criteria.

Inclusion Criteria	Exclusion Criteria
Studies published after 2006	Studies published before 2006
Studies with *P. micra* and bacteremia in the title	German, Korean, and other non-English-language studies
Studies in Japanese and English	Studies in which *P. micra* was detected in samples other than blood

**Table 2 healthcare-10-01727-t002:** Studies included in the review.

Source	Age (Years)	Sex	Chief Complaint	Underlying Cause	Medical History	Organism(s)	Antibiotic Treatment	Drainage	Outcome	Infection Route
[11]	65	Male	Fever, chill, weight loss, abdominal pain	Renal abscess	Renal cyst	*P. micra*	MEPN, VCM→ETP, CLDM	-	Survival	Dental treatment→oral infection
[12]	72	Female	Fever, general malaise, dyspnea	Liver abscess, brain abscess	Colorectal cancer, psoriasis, hypertension	*P. micra*	CPFX, MNZ→MEPN	+	Survival	Unknown
[13]	85	Male	Fever, general malaise	Septic pulmonary embolism	Diabetes mellitus	*P. micra*	MEPN, VCM→ABPC/SBT	-	Survival	Periodontitis/oral infection
[14]	94	Male	Fever, constipation	Colorectal cancer	Dyslipidemia, diabetes, hypertension	*P. micra* *Gemella morbillorum*	CTRX→ ABPC/SBT	+	Survival	Unknown
[15]	61	Male	Fever, headache	Bacteremia meningitis	Chronic hepatitis B, dyslipidemia	*P. micra*	CTRX, VCM→ABPC/SBT, MNZ	-	Survival	Dental treatment/oral infection
[16]	53	Male	Fever, general malaise, chills	Esophageal cancer	Chronic alcoholic liver disease	*P. micra*	MEPN	-	Septicemia Death (6 week)	Periodontal disease /oral infection
[17]	85	Male	Fever, chills	Bile duct lithiasis	Hypertension	*P. micra*	CPFX, PCG	-	Survival	Endoscopic procedures/mucosal infection
[18]	43	Female	Fever, abdominal pain, vomiting	Retroperitoneal leiomyosarcoma	None	*Atopobium rimae* *P. micra*	LVFX→TAZ/PIP, MNZ	-	Septicemia Death (20 days later)	Retroperitoneal tumor necrosis/unknown
[19]	82	Male	Fever, dizziness	Infectious endocarditis pacemaker infection	Sinusoidal failure syndrome, hypertension, abdominal aortic aneurysm	*P. micra*	CTRX→ABPC→GM	-	Survival	Dental caries/oral infection
[20]	71	Male	Fever, chills, cough	Infectious endocarditis	Diabetes mellitus	*P. micra*	CTRX, VCM→PCG	-	Survival	Dental treatment/oral infection
[21]	72	Male	Abdominal pain, night sweats, fever	intrahepatic portal vein thrombosis	Hypertension	*P. micra*	LVFX→TAZ/PIP→CPFX, MNZ	-	Survival	Dental treatment/oral infection
[22]	65	Female	Loss of appetite, nausea, weakness	Liver abscess, brain abscess	None	*P. micra*	CTX, MNZ	+	survival	pressure ulcer/transdermal infection
[23]	67	Male	Abdominal pain	Spondylodiscitis	Diabetes mellitus	*P. micra*	ABPC/SBT→ABPC	-	Survival	Periodontal disease, dental treatment/oral infection
[24]	83	Male	Lumbago	Spondylodiscitis, epidural abscess, psoas abscess	Left hip joint replacement, right knee joint replacement, ischemic heart disease	*P. micra*	ABPC, GM→CLDM, REP	-	Survival	Unknown
[25]	83	Male	Lumbago	Epidural abscess	Aortic regurgitation, Atrial fibrillation	*Desulfovibro fairfieldensis* *P. micra*	ETP→PCG	+	Survival	Bacterial translocation
[26]	77	Male	Fever, abdominal pain, left shoulder pain, shortness of breath, weight loss	Splenic vein thrombosis, splenic abscess	Jejunal perforation	*P. micra*	TAZ/PIP, VCM→ABPC/SBT→ABPC	+	Survival	Unknown
[27]	42	Male	Fever, chills, cough	Infectious endocarditis	MVR, diabetes mellitus	*P. micra*	CTX, VCM	-	Survival	Dental treatment/oral infection
[28]	82	Male	Back pain, shoulder pain	Spondylodiscitis, psoas abscess, epidural abscess	Renal failure, gout, lumbar discectomy	*P. micra*	ABPC, MEPM	+	AAA rupture death	Periodontal disease/oral infection
[28]	72	Male	Lumbago	Spondylodiscitis, epidural abscess	Parkinson disease	*P. micra*	ABPC/CVA	+	Survival	Dental treatment/oral infection
[28]	72	Male	Lumbago	Degenerative changes L4/5	Diabetes mellitus	*P. micra*	PCG, ETP→CLDM	-	Survival	Unknown
[28]	72	Female	Lumbago	Spondylodiscitis, epidural abscess	Metastatic breast cancer	*P. micra*	ABPC/CVA→MFLX	+	Survival	Unknown
[4]	73	Male	Pain in right knee	Septic arthritis of the knee, pseudogout	Hypertension, diabetes mellitus, total hip replacement	*Staphylococcus aureus* *P. micra*	VCM→PCG	+	Survival	
[29]	41	Female	Abdominal pain, tachycardia, low blood pressure	Ulcerative colitis, pulmonary embolism	Ulcerative colitis, deep vein thrombosis	*Clostridium cadaveris*, *P. micra*	TAZ/PIP→LZD, MNZ, MEPN, FCZ	-	Septicemia Death (55 days later)	
[30]	83	Male	Lumbago	Spondylodiscitis, psoas abscess, necrotizing fasciitis	Prostate cancer, diabetes mellitus, hypertension, hyperuricemia, lumbar compression fracture	*P. micra*	MEPN, CLDM	+	Septicemia Death (3 days later)	Diabetes, prostate cancer/unknown
[31]	70	Male	Chest pain	Infectious endocarditis	None	*P. micra*	ABPC/SBT	+	Survival	Dental caries/oral infection
[32]	59	Male	Lumbago	Spondylodiscitis	Lumbar discectomy	*P. micra*	ETP→ABPC	+	Survival	Periodontal disease/oral infection

Abbreviations: ABPC, ampicillin; ABPC/SBT, sulbactam/ampicillin; AMPC/CVA, clavulanic acid/amoxicillin; CLDM, clindamycin; CPFX, ciprofloxacin; CTRX, ceftriaxone; CTX, cefotaxime; ETP, ertapenem; FCZ, fluconazole; GM, gentamicin; LVFX, levofloxacin; LZD, linezolid; MEPN, meropenem; MNZ, metronidazole; PCG, penicillin G; TAZ/PIPC, tazobactam/piperacillin; REP, rifampicin; VCM, vancomycin.

## Data Availability

All relevant data are presented in the manuscript.

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
