# Peer review of "Infection Route of Parvimonas micra: A Case Report and Systematic Review"

_healthcare, 2022, doi:10.3390/healthcare10091727_

Round 1

Reviewer 1 Report

I  cant undertstand rationale of case report with systematic review.

What is new in your case report , 

This is just infection reported.

Authors need to revisit systemtatic review protocol.

Your review is just record of 32 studies with clinical features

Where is Forest plot ?

Author Response

Our responses to the Reviewers’ comments

Thank you for reviewing our manuscript and providing suggestions for its improvement. We have provided point-by-point responses to the Reviewers’ comments. Our revisions are indicated in red font here and in the manuscript. We hope that the revised manuscript meets the journal’s requirements and can now be considered for publication.

  cant undertstand rationale of case report with systematic review.

 Response: Thank you for your valuable feedback. Based on your comment, we have added the rationale for this case report and for undertaking a systematic review of case reports in the Introduction section, as follows:

“It remains unknown how P. micra monoinfections are transmitted and which patient populations are affected. Clarification of the transmission of P. micra monoinfections is vital for prevention and effective treatment. When P. micra is detected in blood cultures, early diagnosis and therapeutic interventions are possible. If the most common routes of infection could be identified, along with the background of patients susceptible to P. micra bacteremia, clinicians could more accurately determine a targeted treatment pathway against the primary disease. However, clinical diagnosis of P. micra monoinfections and treatment remain challenging. Here, we present a 54-year-old man with P. micra bacteremia and aimed to identify the source of the invasion and its clinical features, to highlight challenges in diagnosis and treatment. We also aimed to investigate the transmission of P. micramonoinfections and identify the affected patient population through undertaking a systematic review of relevant case reports concerning P. micra bacteremia.” (page 2, lines 68-80)

What is new in your case report , 

 Response: Thank you for your valuable feedback. We have added the following text to the Introduction section.

“When P. micra is detected in blood cultures, early diagnosis and therapeutic interventions are possible. If the most common routes of infection could be identified, along with the background of patients susceptible to P. micra bacteremia, clinicians could more accurately determine a targeted treatment pathway against the primary disease. However, clinical diagnosis of P. micramonoinfections and treatment remain challenging. Here, we present a 54-year-old man with P. micra bacteremia and aimed to identify the source of the invasion and its clinical features, to highlight challenges in diagnosis and treatment.” (page 2, lines 70-77)

 This is just infection reported.

Authors need to revisit systemtatic review protocol.

Your review is just record of 32 studies with clinical features

 Response: Thank you for your valuable feedback. We have added modified our description of the protocol for our systematic review in the Materials and Methods section, as follows:

2.2. Systematic review

2.2.1. Search strategy

The following keywords were searched on Google Scholar and PubMed online databases: “Parvimonas micra” and “bacteremia”. (page 3, lines 107-110)

2.2.2. Inclusion and exclusion criteria

Studies written in Japanese and English and published after 2006 on bacteremia associated with P. micra and associated diseases, such as sepsis, abscess, and spondylitis, were included. Studies published prior to 2006, those written in languages other than Japanese or English, and those in which P. micra was detected in samples other than blood were excluded. Inclusion and exclusion criteria are summarized in Table 1. (page 3, lines 112-117)

2.2.3. Data extraction

We used the search terms “Parvimonas micra” and “bacteremia” in Google scholar and PubMed online databases. The first and second authors read the full text and independently extracted the data. After data extraction, they discussed the extracted data and, once consensus had been reached regarding the inclusion of a study, the data were captured in an Excel file. This systematic review was registered in PROSPERO (registration number: 316707) and conducted according to standard procedures..” (page 3, lines 121-127)

Where is Forest plot ?

Response: Thank you for your question. We undertook a systematic review but did perform a meta-analysis; therefore, we did not include a Forest plot.

Reviewer 2 Report

The authors have evaluated a case of a 54-year-old man with P. micra bacteremia to identify the source of the invasion and its clinical features as well as the results of a systematic review of cases of P. micra bacteremia, in this study.

Plagiarism checked, it is suitable for ethical situaiton.

Reference’s quality: Current references are used.

Comment: The article is overall well written, very informative and will contribute a lot to the literature. However, references need to be reviewed again. For example, is the 36th reference correct ? I could not find this reference from google scholar and pubmed. The discussion section says Dahl et al. on the 167th line, but ‘’Sakamoto, H.’’ is written in the reference, this situation must be corrected.

Evaluation report: Congratulations to the authors for a good work, this study overall well written.  Abstract, Introduction, materials and methods, results and discussion sections of the article have enaugh scientific knowledge. I think it is acceptaple after minör revisions.

Author Response

Our responses to the Reviewers’ comments

Thank you for reviewing our manuscript and providing suggestions for its improvement. We have provided point-by-point responses to the Reviewers’ comments. Our revisions are indicated in red font here and in the manuscript. We hope that the revised manuscript meets the journal’s requirements and can now be considered for publication.

The authors have evaluated a case of a 54-year-old man with P. micra bacteremia to identify the source of the invasion and its clinical features as well as the results of a systematic review of cases of P. micra bacteremia, in this study. 

Plagiarism checked, it is suitable for ethical situaiton.

 Response: Thank you for your valuable feedback.

Reference’s quality: Current references are used.

Response: Thank you for your valuable feedback.

Comment: The article is overall well written, very informative and will contribute a lot to the literature. However, references need to be reviewed again. For example, is the 36th reference correct ? I could not find this reference from google scholar and pubmed. The discussion section says Dahl et al. on the 167th line, but ‘’Sakamoto, H.’’ is written in the reference, this situation must be corrected.

Response: Thank you for highlighting these matters. As per your comment, we have reviewed the references and the main manuscript and have revised the references and all intext citations and descriptions accordingly.

Evaluation report: Congratulations to the authors for a good work, this study overall well written.  Abstract, Introduction, materials and methods, results and discussion sections of the article have enaugh scientific knowledge. I think it is acceptaple after minör revisions.

Response: Thank you for your positive feedback.

Reviewer 3 Report

Dear author, First, I congratulate you for this work. The writing style is very good and the objective was well presented. I have some short comments that may improve the discussion of this paper.

1. I felt a lack of argumentation about the social factors that may be related to the dynamics of P. micra. Is there any supposed association with education level? Ocuppational level? Any specific habits? I think that other information about the epidemiology of P. micra (not only related to medical and/or biological context) will be very welcome. In addition, such social factors do not only need be available by scientific literature, but also may be described according to your own experience.

2. I suggest you make the letter size of Figure 1 a little bit larger.

3. Since the authors are from Japan, I also believe it could be interesting to describe something about the scenery of P. micra epidemiology in your country (if possible).

Thank you for attention.

Author Response

Our responses to the Reviewers’ comments

Thank you for reviewing our manuscript and providing suggestions for its improvement. We have provided point-by-point responses to the Reviewers’ comments. Our revisions are indicated in red font here and in the manuscript. We hope that the revised manuscript meets the journal’s requirements and can now be considered for publication.

Dear author, First, I congratulate you for this work. The writing style is very good and the objective was well presented. I have some short comments that may improve the discussion of this paper.

  1. I felt a lack of argumentation about the social factors that may be related to the dynamics of P. micra. Is there any supposed association with education level? Ocuppational level? Any specific habits? I think that other information about the epidemiology of P. micra (not only related to medical and/or biological context) will be very welcome. In addition, such social factors do not only need be available by scientific literature, but also may be described according to your own experience.

Response: Thank you for your valuable and encouraging feedback. Per your comment, we have added the descriptions regarding social and educational factors in relation to this infection as follows.

“The pathogenicity of P. micra includes the presence of capsid, high protease activity and hydrogen sulfide toxicity produced by the utilization of glutathione [3]. Socially isolated patients, those with a low socioeconomic status, those with poor hygiene, and immunocompromised patients are vulnerable to this infection and may be at high risk owing to the absence of habitual oral care [2,3].” (page 2, lines 46-50)

  1. I suggest you make the letter size of Figure 1 a little bit larger.

 Response: Thank you for your valuable suggestion. We have revised the scale of Figure 1 to enhance readability.

  1. Since the authors are from Japan, I also believe it could be interesting to describe something about the scenery of P. micra epidemiology in your country (if possible).

 Response: Thank you for your valuable feedback. We have added descriptions concerning the prevalence in the Discussion section, as follows.

“All patients who presented with infective endocarditis had a history of dental procedures, such as tooth extractions. It has long been known that transient bacteremia due to dental procedures is a cause of infective endocarditis [34]. Bacteria entering the bloodstream after dental procedures, such as tooth extraction, are rapidly eliminated by the liver and other reticuloendothelial tissues, and most disappear from the bloodstream within a short period; this is referred to as transient bacteremia [35]. Sakamoto [36] reported that the frequency of bacteremia after tooth extractions was 100%, with wisdom tooth extractions accounting for 55%, and that the frequency of bacteremia after tartar removal was 70%. The frequency of bacteremia has been reported to be higher for periodontal extractions [37]. Non-dental bacteremia has been reported as due to daily tooth brushing (30%), chewing (38%), and dental flossing [38–44]. Of these, the risk associated with tooth brushing has been emphasized in recent years [38,39]. The presence of bacteremia induced in the oral cavity through such routine practices has led to a strong recognition of the importance of routine oral follow-ups [39,45]. The European Society of Cardiology guidelines recommend that, as a general precaution, high- and intermediate-risk patients should undergo rigorous dental follow-up twice a year for those at particularly high risk and once a year for others. To prevent infectious endocarditis, control of periodontal disease and caries is critical, and regular dental and oral management is crucial, even when a patient is asymptomatic.” (page 12, lines 198-216)

Thank you for your attention.

Reviewer 4 Report

This is a case-report and systematic review of the infection route of Parvimonas micra, a bacterium of the gastrointestinal tract often found in abscesses and periodontitis as a complex and deep-seated infection such as areas around artificial joints. The paper describes the rare event of a monobacterial bacteremia and aim at identifying its source. Also, a systematic review of 26 monobacteremia infections of P. micra. The major risk factors that were detected are: malignancy, diabetes mellitus, and post-arthroplasty.

Major comments:

·         The introduction is a little too short.

·         Line 91: "Search strategy" – if the name of the bacterium had changed, why not look for its synonyms? I saw you included 2 papers where the Micromonas synonym was used – are you sure you found all relevant papers?

·         Line 138 – "Figure 21. of the 26 patients (81.8%) survived" – I don’t see a figure attached? And there is probably a truncation of the legend as this can't be figure 21

·         Table 2 – please change the font so that the entire headline is not truncated [in chief complaint for example).

·         Lines 147-148 – when you say common – how common? What percent? How many people out of the 26 cases?

·         Line 179 – why is it crucial for people in good physical condition to keep dental hygiene and not for people with moderate or bad condition?

·         Line 187-188: " …Other backgrounds of patients included type 2…" – please cite the papers.

·         Line 191-192: – please cite the papers here too.  

·         Line 207: "..patients did not respond to treatment with metronidazole;" – please add citation here – if it is no. 63 it still should be cited before the ";"

·         Line 217: again please add citation!

·         Line 230 – what do you mean by "it is likely" – do you know or are you speculating? It seems you have no data to support this comment.

Minor comments:

·         Line 33: " obligate anaerobe and a small (0.3–0.7 μm) Gram-positive anaerobic coccus" – please use anaerobe only once, it is redundant to mention this twice.

·         Line 51: " . P. micra may be multi-drug-resistant" – what drugs is it resistant to? Are the citations relevant enough to this claim? I am surprised that papers like this have not been cited - https://www.ncbi.nlm.nih.gov/pmc/articles/PMC7602954/ [I am not connect to this paper in anyway, but I feel it is relevant]

·         Line 56 – the 'However' seems out of place.

Author Response

Our responses to the Reviewers’ comments

Thank you for reviewing our manuscript and providing suggestions for its improvement. We have provided point-by-point responses to the Reviewers’ comments. Our revisions are indicated in red font here and in the manuscript. We hope that the revised manuscript meets the journal’s requirements and can now be considered for publication.

This is a case-report and systematic review of the infection route of Parvimonas micra, a bacterium of the gastrointestinal tract often found in abscesses and periodontitis as a complex and deep-seated infection such as areas around artificial joints. The paper describes the rare event of a monobacterial bacteremia and aim at identifying its source. Also, a systematic review of 26 monobacteremia infections of P. micra. The major risk factors that were detected are: malignancy, diabetes mellitus, and post-arthroplasty.

Major comments:

  • The introduction is a little too short.

Response: Thank you for your valuable feedback. Per your comment, we have added further text concerning P. micra infections, including the prevalence, social factors, and the significance of this case and those identified in our systematic review, as follows:

Parvimonas micra (P. micra) inhabits the oral cavity and intestinal tracts in humans [1]. It is an obligate anaerobe and a small (0.3–0.7 μm) Gram-positive coccus, causing various infections in immunocompromised hosts. Originally, it was identified as Peptostreptococcus micros in 1933, but was reclassified as Micrcomonas micros in 1999, and finally as P. micra in 2006 [2]. P. micra infections have increased in nosocomial and immunocompromised hosts because of aging societies [2]. Primary care healthcare workers need to be aware of these types of infections.

  1. micra is a component of the oral, upper respiratory, and intestinal microflora, and is of major clinical and bacteriological importance owing to its high isolation rate within clinical material owing to poor hygiene among immunocompromised older patients in aging societies [2]. Its pathogenic potential has been implicated in chronic periodontal disease, alveolar abscess, peritonsillar abscess, chronic sinusitis, chronic otitis media, and pulmonary pyogenic disease [3]. It is known to be involved in deep-seated infections, such as those occurring around artificial joints [3]. The pathogenicity of P. micra includes the presence of capsid, high protease activity and hydrogen sulfide toxicity produced by the utilization of glutathione [3]. Socially isolated patients, those with a low socioeconomic status, those with poor hygiene, and immunocompromised patients are vulnerable to this infection and may be at high risk owing to the absence of habitual oral care [2,3].
  2. micra is difficult to identify because of its lack of clinical symptoms, slow growth, and the need for special culture media and identification methods [4–6]. Pyogenic spondylitis is the most common infection caused by P. micra, and pyogenic arthritis (post-knee arthroplasty), infective endocarditis, pleurisy, meningitis, and brain abscess have been reported [7]. As P. micra is an anaerobic inhabitant of the oral microbiome, risk factors include dental procedures, periodontitis, tooth extractions, and oral infections, such as abscesses or caries on the lingual apex [7]. P. micra may be multi-drug-resistant and may co-infect with other multi-drug-resistant bacteria in relation to cephalosporins and quinolone, resulting in a polymicrobial etiology for endogenous oral infections, such as periodontitis, while also being detected in soft tissues, skin infections, and various abscesses [8–11]. P. micraco-pathogens associated with polymicrobial infections include Streptococcus, Bacteroides, and Fusobacterium [12]. In increasingly aging societies, healthcare workers encounter various symptoms in older patients, and the etiologies of their symptoms could derive from the bacteremia of P. micra. Therefore, healthcare workers should be mindful of the likelihood of P. micra infections among older adults.

It remains unknown how P. micra monoinfections are transmitted and which patient populations are affected. Clarification of the transmission of P. micra monoinfections is vital for prevention and effective treatment. When P. micra is detected in blood cultures, early diagnosis and therapeutic interventions are possible. If the most common routes of infection could be identified, along with the background of patients susceptible to P. micra bacteremia, clinicians could more accurately determine a targeted treatment pathway against the primary disease. However, clinical diagnosis of P. micra monoinfections and treatment remain challenging. Here, we present a 54-year-old man with P. micra bacteremia and aimed to identify the source of the invasion and its clinical features, to highlight challenges in diagnosis and treatment. We also aimed to investigate the transmission of P. micramonoinfections and identify the affected patient population through undertaking a systematic review of relevant case reports concerning P. micra bacteremia.” (Introduction, page 1)

  • Line 91: "Search strategy" – if the name of the bacterium had changed, why not look for its synonyms? I saw you included 2 papers where the Micromonas synonym was used – are you sure you found all relevant papers?

Response: Thank you for your valuable feedback. We checked our search strategy and we have also now included articles with ‘Parvimonas micra’.

  • Line 138 – "Figure 21. of the 26 patients (81.8%) survived" – I don’t see a figure attached? And there is probably a truncation of the legend as this can't be figure 21

Response: Thank you for highlighting this error. We have revised this typographical error in the manuscript.

  • Table 2 – please change the font so that the entire headline is not truncated [in chief complaint for example).

Response: Thank you for your valuable suggestion. We have revised the font and added a separating line under the headlines.

  • Lines 147-148 – when you say common – how common? What percent? How many people out of the 26 cases?

Response: Thank you for these important questions. We have added an explanation of the frequency of each description by means of including the number of the cases.

  • Line 179 – why is it crucial for people in good physical condition to keep dental hygiene and not for people with moderate or bad condition?

Response: Thank you for this important question. The relevant paragraph has been revised, as follows:

“The European Society of Cardiology guidelines recommend that, as a general precaution, high- and intermediate-risk patients should undergo rigorous dental follow-up twice a year for those at particularly high risk and once a year for others. To prevent infectious endocarditis, control of periodontal disease and caries is critical, and regular dental and oral management is crucial, even when a patient is asymptomatic.” (page 12, lines 211-216)

  • Line 187-188: " …Other backgrounds of patients included type 2…" – please cite the papers.

Response: Thank you for your valuable feedback. We have added the relevant references and revised the descriptions, as follows:

“In this study, other relevant patient histories included type 2 diabetes mellitus, colorectal cancer, pressure ulcers, and aspiration. Previous studies have reported that P. micra bacteremia is associated with aspiration pneumonia, suggesting that postoperative stress, as well as increased dead space due to increased sputum production, may have led to an increase in P. micra [16,17,19].(page 12, line 224-page 13, line 229)

  • Line 191-192: – please cite the papers here too.  

Response: Thank you for your valuable feedback. Per your comment, we have added the references and revised the descriptions as follows.

“In addition, some cases of P. micra bacteremia possibly due to ulcers or endoscopic procedures have been reported, and trauma due to intestinal mucosal injury may predispose to bacteremia. Nine cases had no description of the route of infection in the case presentation or discussion, with the patients presenting with liver abscess, splenic abscess, spinal discitis, epidural abscess, intradural abscess, and diabetes mellitus, and no history of dental treatment.” (page 13, lines 229-234)

  • Line 207: "..patients did not respond to treatment with metronidazole;" – please add citation here – if it is no. 63 it still should be cited before the ";"

Response: Thank you for your valuable feedback. We have added the reference and revised the text, as follows:

“Concerning treatment and antimicrobial susceptibility, in one previous study, 3.2% of patients did not respond to treatment with metronidazole [63]; however, most patients did respond to treatment with common antimicrobial agents.” (page 13, lines 247-251)

  • Line 217: again please add citation!

Response: Thank you for highlighting this omission. The relevant references have been added and we have revised the text accordingly, as follows.

“Regarding prognosis, one patient with in situ carcinoma became debilitated owing to the infection and ruptured an abdominal aortic aneurysm, which ultimately resulted in death [57-59]. Anaerobic infections such as P. micra require early detection and treatment [53-56]. In patients who are at risk of severe disease, the threshold for searching for infections such as abscesses and infective endocarditis should be lowered for those with diabetes mellitus, non-invasive cancer, and poor oral hygiene, even in those with non-specific symptoms such as fever and malaise [60-62]. Abscess formation should be considered, especially when accompanied with localized pain, as P. micra is an anaerobe and prone to abscess formation.” (page 13, lines 260-268)

  • Line 230 – what do you mean by "it is likely" – do you know or are you speculating? It seems you have no data to support this comment.

 Response: Thank you for your valuable question. We have revised the text and have added further descriptions concerning consideration of the clinical course and the results of this research, as follows:

“Our patient was treated in the emergency room, and the details of the disease course remain unknown. Our patient had a history of dental treatment, oral contamination, a painful mouth, and no other febrile findings, suggesting a high probability that P. micra bacteremia occurred via the oral cavity. Our patient was an immunocompetent individual with no history of diabetes mellitus, cancer, or steroid use. Based on the clinical course, the patient had transient bacteremia caused by P. micra that resolved spontaneously. Therefore, consistent follow-up and health maintenance are needed because of the possibility of abscess formation, infective endocarditis, or spondylitis.” (page 13, lines 270-277)

Minor comments:

Line 33: " obligate anaerobe and a small (0.3–0.7 μm) Gram-positive anaerobic coccus" – please use anaerobe only once, it is redundant to mention this twice.

Response: Thank you for pointing this out. We have deleted the redundant word.

 Line 51: " . P. micra may be multi-drug-resistant" – what drugs is it resistant to? Are the citations relevant enough to this claim? I am surprised that papers like this have not been cited - https://www.ncbi.nlm.nih.gov/pmc/articles/PMC7602954/ [I am not connect to this paper in anyway, but I feel it is relevant]

Response: Thank you for your valuable questions. We have added the relevant names of the antibiotics.

Line 56 – the 'However' seems out of place.

Response: Thank you for your valuable feedback. ‘However’ has been deleted accordingly.

Round 2

Reviewer 1 Report

Logic for using case report with systematic review